# WAVEFORMER: LINEAR-TIME ATTENTION WITH FORWARD AND BACKWARD WAVELET TRANSFORM

## ABSTRACT

We propose Waveformer that learns attention mechanism in the wavelet coefficient space, requires only linear time complexity, and enjoys universal approximating power. Specifically, we first apply *forward* wavelet transform to project the input sequences to multi-resolution orthogonal wavelet bases, then conduct non-linear transformations (in this case, a random feature kernel) in the wavelet coefficient space, and finally reconstruct the representation in input space via *backward* wavelet transform. We note that other non-linear transformations may be used, hence we name the learning paradigm Wavelet transformation for Sequence lEarning (WISE). We emphasize the importance of backward reconstruction in the WISE paradigm — without it, one would be mixing information from both the input space and coefficient space through skip-connections, which shall not be considered as mathematically sound. Compared with Fourier transform in recent works, wavelet transform is more efficient in time complexity and better captures local and positional information; we further support this through our ablation studies. Extensive experiments on seven long-range understanding datasets from the Long Range Arena benchmark and code understanding tasks demonstrate that (1) Waveformer achieves competitive and even better accuracy than a number of state-of-the-art Transformer variants and (2) WISE can boost accuracies of various attention approximation methods without increasing the time complexity. These together showcase the superiority of learning attention in a wavelet coefficient space over the input space.

## 1 INTRODUCTION

Transformer (Vaswani et al., 2017) has become one of the most influential models in natural language processing (Devlin et al., 2018; Brown et al., 2020), computer vision (Dosovitskiy et al., 2020), speech processing (Baevski et al., 2020), code understanding (Chen et al., 2021a) and many other applications. It is composed of the attention layer and the feed-forward layer with layer norms and skip-connections added in between. The original design of the attention layer scales quadratically to the sequence length, becoming a scalability bottleneck of Transformers as texts, images, speech, and codes can be of vast lengths.

State-of-the-art attention approximation methods have enabled Transformers to scale sub-quadratic or even linearly to the input sequence length. Typical approaches to computing a cheaper pseudo-attention include sparse attention patterns (Parmar et al., 2018; Wang et al., 2019; Beltagy et al., 2020; Zaheer et al., 2020), low-rank approximation (Wang et al., 2020; Chen et al., 2021b), and kernel approximation (Katharopoulos et al., 2020; Choromanski et al., 2020; Peng et al., 2020), where most of these methods are linear in time complexity. For a comprehensive review, please refer to Section 4.

Recent works on improving the effectiveness and efficiency of long-range capabilities of Transformers start to explore attention learning in a transformed space. For example, conducting low-cost token-mixing with forward Fourier transform leads to remarkable accuracy improvement with a quasi-linear time complexity (Lee-Thorp et al., 2021). Token-mixing ideas (You et al., 2020; Lee-Thorp et al., 2021) are simple and effective, however, they lose Transformer's universal approximating power by replacing attention with hard averaging (Yun et al., 2019). Moreover, without backward transform the model will mix information from both the input and transformed spaces,

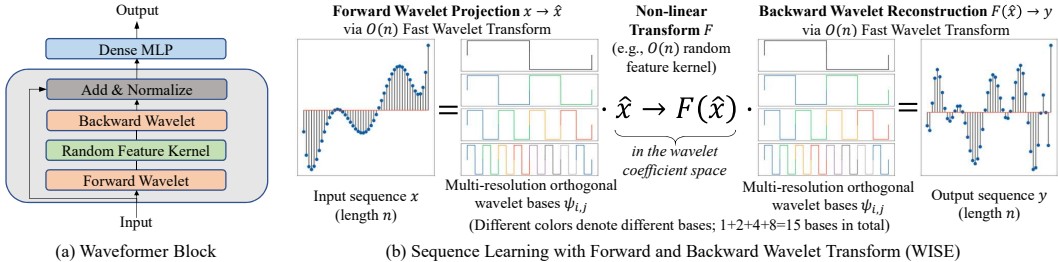

Figure 1: An overview of our proposed Waveformer and WISE. (a) The only difference between a Transformer block and a Waveformer block is the attention computation. (b) The general flow of computation in WISE with forward and backward wavelet transform.

which is not mathematically sound. Since multiplication in the Fourier coefficient space, after projected back to the input space, is equivalent to directly calculating convolutions in the input space, people have also utilized the forward and backward Fourier transform to learn large global filters with linear weights (Rao et al., 2021) and non-linearities (Guibas et al., 2021).

We propose Waveformer that facilitates the attention mechanism learning in a *wavelet coefficient space*, as shown in Figure 1(a). It requires only *linear time complexity* and enjoys *universal approximating power*. Specifically, we first apply *forward* wavelet transform to project the input sequence to multi-resolution orthogonal wavelet bases, then conduct non-linearity (e.g., random feature kernel (Rahimi & Recht, 2007)) in the wavelet coefficient space, and finally, reconstruct the representation in input space via *backward* wavelet transform. We name this general learning paradigm WISE, as shown in Figure 1(b), it can be suited with attention approximation methods to boost their long-range understanding capabilities. We implement wavelet transform using Fast Wavelet Transform (FWT) (Mallat, 1989) so both transform steps are linear in time. Intuitively, WISE operates on a local to global, coarse to fine-grained cascading structure. Compared with Fourier transform, wavelet transform is more efficient in time complexity and better captures local and positional information since the wavelet basis is localized in space with ranging granularity. For the non-linear transformation in the wavelet coefficient space, one can apply any non-linearities, while we suggest using attention approximation methods. A reason behind this is that since wavelet transformation is invertible and exact, WISE will be universal approximating when coupled with universal approximators as its non-linearity.

We conduct extensive experiments on the Long Range Arena (LRA) benchmark and common code understanding tasks to empirically ablate and justify this method. Compared with a number of widely-used Transformer variants, Waveformer with a linear time complexity can achieve competitive and even better performance. When combined with various representative attention approximation functions, WISE can boost their performance without incurring extra time complexities. This shows that learning in a wavelet coefficient space provides better long-range understanding capability over direct learning in the input space. Our ablation studies also support the use of the forward-backward schema and the superiority of wavelet transform over Fourier transform.

In summary, our major contributions are as follows.

- We propose WISE to facilitate learning in the wavelet coefficient space following a forward-backward paradigm which can be suited with attention approximation methods while boosting their long-range understanding capabilities.
- Based on WISE, we develop Waveformer that requires only linear time complexity and enjoys universal approximating power for sequence-to-sequence functions.
- Extensive experiments on the Long-Range Arena benchmark and code understanding tasks have demonstrated the effectiveness and also justified the design of Waveformer.

**Reproducibility.** We will release our code on GitHub.

## 2 WAVEFORMER

As shown in Figure 1(a), the only difference between a Transformer block and a Waveformer block is the attention computation. In this section, we introduce the details that replace the attention

computation in Waveformer. The general flow of WISE is shown in Figure 1(b), which constitutes the forward wavelet transform, the non-linearity in the middle, and the backward wavelet transform.

We list our notations here — we denote scalars as $x$, vectors as $\mathbf{x}$, matrices as $X$; we denote inner product between functions $f, g$ as $\langle f, g \rangle = \int f(t)g(t) \, \mathrm{d}t$; we denote function $f$'s transformation in the coefficient space as $\hat{f}$.

**Background about Attention**    Let $X \in \mathbb{R}^{n \times d}$ denotes the input sequence of length $n$ and hidden dimension $d$. A dense self-attention layer is shown below:

$$\text{Attention}(X) = \text{Softmax}(\frac{QK^\top}{\sqrt{d}})V \tag{1}$$

where $Q = XW_q$, $K = XW_k$, $V = XW_v$ with $W_q, W_k, W_v \in \mathbb{R}^{d \times m}$ stand for the query, key, and value, respectively. The attention head size is denoted by $m$. The self-attention layer $\text{Attention}(\cdot)$ computes a weighted average for each column of $V$ according to the dot-product similarity $QK^\top$ that is of $O(mn^2)$ number of operations.

## 2.1 Generalized Forward-Backward Paradigm

We propose the generalized forward-backward paradigm to conduct learning in the coefficient space between forward and backward transformation. WISE is a special case of the generalized paradigm when the transformation is wavelet transform. The forward transformation, also called analysis, decomposes the input sequence into coefficients of a set of orthogonal, complete functional basis. We then conduct non-linear transformation in the coefficient space, which directly operates on the function. In the backward transformation, also called synthesis, we reconstruct the target representation in the original function space.

We require the forward-backward transformation pair to be invertible and exact, meaning that one can perfectly reconstruct the same input from the coefficients. Note that orthogonality is not necessary for this condition but a non-orthogonal basis cannot be transformed via simple inner product.

The general framework is shown below. Without loss of generality, we limit ourselves to 1d functions. Given input and output function $x(t), y(t) : \mathbb{R} \to \mathbb{R}$ on time domain $t$, functional basis $g(\omega, t)$ on both time domain $t$ and frequency domain $\omega$, and non-linear transformation $F$.

$$\text{Forward Transform}: \quad \hat{x}(\omega) = \int x(t)g^*(\omega, t) \, \mathrm{d}t = \sum_i x(t_i)g^*(\omega, t_i) \tag{2}$$

$$\text{Nonlinear Transform}: \quad \hat{h}(\omega) = F \circ \hat{x}(\omega) \tag{3}$$

$$\text{Backward Transform}: \quad y(t) = \int \hat{h}(\omega)g(\omega, t) \, \mathrm{d}\omega = \sum_j \hat{h}(\omega_j)g(\omega_j, t) \tag{4}$$

where $g^*(\omega, t)$ denotes the complex conjugate of $g$ in case $g$ has complex parts.

As a concrete example, we use the Fourier transformation pair (formally defined in Appendix A.2) to illustrate this idea. Under Fourier transform, the functions are decomposed into sinusoidal waves, with $g(\omega, t) = e^{i2\pi\omega t}$. The coefficients will represent the magnitude of the corresponding sinusoidal function. Hence, learning the mapping between any sequential function $x(t) \to y(t)$ becomes learning a coefficient mapping between the transformed $\hat{x}(\omega) \to \hat{y}(\omega)$ in a vector space.

In WISE, we utilize the more localized wavelet transform as the forward-backward mechanism and in Waveformer we use the random feature kernel as the non-linear transformation. However we want to highlight that numerous alternative candidates can fit in our generalized paradigm, we briefly list a few below. For the forward-backward mechanism, any transformation based on orthogonal polynomials such as Chebyshev transform and Hartley transform will suit our purposes. For the non-linear transformation, essentially any neural architecture with sufficiently strong learning power will be a valid candidate.

We will explain the inner workings of our Waveformer's forward-backward transformation and the reason for choosing it in the next part.

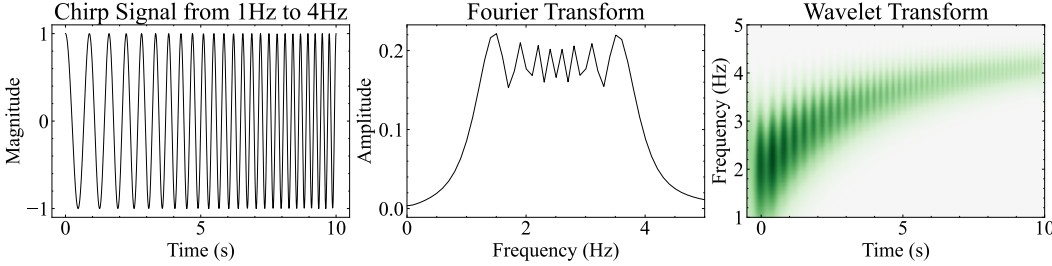

Figure 2: We show a chirp signal from 1Hz to 4Hz on the left, its Fourier transform in the middle and its wavelet transform on the right. As you can see, from the Fourier transform graph one can only infer the existence of signal in the range of 1-4Hz without information on its happening time, while in the wavelet transform graph both time and frequency information are present and one can tell this is a chirp signal.

## 2.2 WAVELET TRANSFORM

Fourier transform decomposes the entire function into global sinusoidal waves. It tells people what *frequencies* are there and in what *magnitude*, but no information is given about *when* that frequency started or ended. See Figure 2 for an illustration on a chirp signal. This limits the capability to understand the local structures of the input and to conduct learning on top of it, which is crucial to many machine learning tasks.

Wavelet transform is designed to solve this issue. It employs a function $\psi(x), x \in \mathbb{R}$, called mother wavelet, to generate a family of translated and dilated wavelets (see Figure 1(b)):

$$\psi_{i,j}(x) = 2^{\frac{i}{2}}\psi(2^i x - j), \quad i, j \in \mathbb{Z} \tag{5}$$

where scale $i$ controls the resolution of the wavelet and $j$ controls the position of the wavelet. With a larger $i$ the wavelet will be squeezed shorter in space, hence the normalization factor $2^{\frac{i}{2}}$ to ensure the same $L^2$ norm for all wavelets. The wavelet family $\psi_{i,j}(x)$ is orthogonal on this dyadic grid.

To be a valid mother wavelet $\psi(x)$, the only requirement is admissibility:

$$\int_{\mathbb{R}} \psi(x)\, \mathrm{d}x = 0 \tag{6}$$

In other words, the sum of function value should be 0.

Given any square integrable function $f \in \mathbb{L}^2(\mathbb{R})$ (i.e., $\int |f(x)|^2\, \mathrm{d}x < \infty$) and wavelet functions $\psi_{i,j}$, the wavelet transform pair is defined as:

$$\text{Forward Discrete Wavelet}: \quad \hat{f}(i,j) = \int_{\mathbb{R}} f(x)\psi_{i,j}^*(x)\, \mathrm{d}x = \sum_t f(x_t)\psi_{i,j}^*(x_t) \tag{7}$$

$$\text{Backward Discrete Wavelet}: \quad f(x) = \sum_{i=-\infty}^{+\infty} \sum_{j=-\infty}^{+\infty} \hat{f}(i,j)\psi_{i,j}(x) \tag{8}$$

where $\psi_{i,j}^*(x)$ denotes the complex conjugate of $\psi_{i,j}(x)$.

Intuitively, in wavelet transform, we are scanning $f(x)$ with a microscope that has two knobs. One knob is the location $j$, the other one is the frequency (i.e., $2^i$). We will be able to oversee the local structure of the input and calibrate it accordingly with parameterized functions in WISE paradigm. See Figure 2 for a direct comparison.

To generalize beyond $\mathbb{L}^2(\mathbb{R})$ and avoid using an infinite number of wavelets, we must introduce another function $\phi$, called scaling function with a similar admissibility and orthogonality constraint:

$$\int_{-\infty}^{+\infty} \phi(x)\, \mathrm{d}x = 1, \quad \phi_{i,j}(x) = 2^{\frac{i}{2}}\phi(2^i x - j), \quad \text{s.t.} \langle \phi_{i,j}, \psi_{i',j'} \rangle = 0, \ i' > i, \ \forall j, j' \tag{9}$$

$\phi_{i,j}$ is designed to cover the scale up to $i$, hence the orthogonality requirement. The decomposition of $f(x)$ therefore becomes:

$$f(x) = \sum_{j=-\infty}^{+\infty} \langle \phi_{0,j}, f \rangle \phi_{0,j}(x) + \sum_{i=0}^{+\infty} \sum_{j=-\infty}^{+\infty} \langle \psi_{i,j}, f \rangle \psi_{i,j}(x) \tag{10}$$

Note that although $i$ still goes to $+\infty$ in (10), $i$ usually has an upper limit in practice since it is impossible to work with infinite frequency. We also highlight that both the forward and backward discrete wavelet transform have an efficient $O(n)$ complexity algorithm (Mallat, 1989).

In the $d$-dimensional case, we do not have a general orthogonal discrete $\mathbb{R}^d$ wavelet, unlike the continuous case. However, we can still perform discrete wavelet transform over each spatial dimension of the input, and we'd still be able to perfectly project and reconstruct the original function.

To summarize, wavelet transform enjoys $O(n)$ time complexity, an already desirable property compared to Fourier transform's $O(n \log n)$ complexity. It further provides the capability to examine the local structures with different resolutions via altering the scale. Learning carried out in this wavelet space will correspond to gathering and processing information from local to global level, in a coarse to fine-grained fashion.

## 2.3 RANDOM FEATURE KERNEL

We have explained the transformation mechanism for WISE. One question remains unanswered: which architecture should we choose to use as the parameterized non-linear transformation? For our purpose, the architecture should have linear complexity in sequence length; should be able to tackle sequential inputs effectively; should be as expressive as possible. Among the ones that fit our criteria, we choose random feature kernel (Rahimi & Recht, 2007) due to its simplicity, efficiency, and theoretical benefits.

We have shown the self-attention in Equation (1). It can be rewritten using the softmax kernel (SM):

$$\text{Softmax}(QK^\top)_{i,j} = \frac{\text{SM}(Q_i, K_j)}{\sum_{j=1}^{n} \text{SM}(Q_i, K_j)}, \quad \text{SM}(x, y) = \exp\left(x^\top y\right) \tag{11}$$

Random feature kernels aim to approximate $\text{SM}(\cdot, \cdot)$ with randomized mapping $\nu : \mathbb{R}^d \to \mathbb{R}_+^r$.

$$K(x, y) = \mathbb{E}[\nu(x)^\top \nu(y)] \tag{12}$$

We use FAVOR+ from Performer (Choromanski et al., 2020) as the randomized mapping $\nu$:

$$\nu(x) = \frac{\exp\left(-\frac{\|x\|^2}{2}\right)}{\sqrt{m}} (\exp\left(w_1^\top x\right), \dots, \exp\left(w_m^\top x\right)), \quad w_i \sim \mathcal{N}(0, \text{I}_d) \;\; \forall i \in \{1, \dots, m\} \tag{13}$$

where $m$ denotes the number of random features. Similar to FAVOR+, we require $w_1, \dots, w_m$ to be orthogonal to reduce the variance of estimation. However, we impose additional normalization on $x$ beforehand to further stabilize the estimation inspired by Peng et al. (2020). We note that this in-place normalization is important when bounding the MSE error of the random feature kernel, as shown in Appendix A.1 Lemma 1.

## 2.4 UNIVERSAL APPROXIMATION POWER

In this subsection, we show that Waveformer has the same universal approximation power for seq-to-seq functions as Transformer. The goal is to show that for any $f$ in $\mathcal{F}$, $\forall p \in [1, +\infty), \forall \epsilon > 0$, we can find a $\bar{f}$ in the class of Waveformer, such that:

$$d_p(f, \bar{f}) = \left( \int_{\mathbb{R}^{n \times d}} \|f(\mathbf{X}) - \bar{f}(\mathbf{X})\|_p^p \, d\mathbf{X} \right)^{\frac{1}{p}} \leq \epsilon$$

We define the Waveformer class that has positional encoding, $h$ heads, head size $s$, hidden dimension $r$ as $\mathcal{W}^{h,s,r}$ with FAVOR+ kernel as described in Section 2.3.

Table 1: Evaluation results on Long-Range Arena benchmark. We show both the average accuracy (Avg) and average accuracy without Retrieval (Avg (w/r)) since LUNA 256, Nyströmformer, and our Waveformer all use prolonged 20k training steps on Retrieval task.

| Model | ListOps | Text | Retrieval | Image | Pathfinder | Avg | Avg (w/r) |
|---|---|---|---|---|---|---|---|
| Transformer | 36.37 | 64.27 | 57.46 | 42.44 | 71.40 | 54.39 | 53.62 |
| Local Attention | 15.82 | 52.98 | 53.39 | 41.46 | 66.63 | 46.06 | 44.22 |
| Sparse Trans. | 17.07 | 63.58 | 59.59 | 44.24 | 71.71 | 51.24 | 49.15 |
| Longformer | 35.63 | 62.85 | 56.89 | 42.22 | 69.71 | 53.46 | 52.60 |
| Linformer | 35.70 | 53.94 | 52.27 | 38.56 | 76.34 | 51.36 | 51.14 |
| Reformer | 37.27 | 56.10 | 53.40 | 38.07 | 68.50 | 50.67 | 49.99 |
| Sinkhorn Trans. | 33.67 | 61.20 | 53.83 | 41.23 | 67.45 | 51.39 | 50.89 |
| Synthesizer | 36.99 | 61.68 | 54.67 | 41.61 | 69.45 | 52.88 | 52.43 |
| BigBird | 36.05 | 64.02 | 59.29 | 40.83 | 74.87 | 55.01 | 53.94 |
| Linear Trans. | 16.13 | 65.90 | 53.09 | 42.34 | 75.30 | 50.55 | 49.92 |
| Performer | 18.01 | 65.40 | 53.82 | 42.77 | 77.05 | 51.41 | 50.81 |
| FNet | 35.33 | 65.11 | 59.61 | 38.67 | 77.80 | 55.30 | 54.23 |
| LUNA 256 | 37.98 | 65.78 | **79.56** | **47.86** | 78.55 | 61.95 | 57.54 |
| Nyströmformer | 37.15 | 65.52 | **79.56** | 41.58 | 70.94 | 58.95 | 53.80 |
| Waveformer | **38.20** | **75.60** | 78.56 | 42.98 | **79.17** | **62.90** | **58.99** |

**Theorem 1** $\forall p \in [1, +\infty)$, $\epsilon > 0$, and for any $f \in \mathcal{F}$, we can find a Waveformer network $\mathsf{w} \in \mathcal{W}^{2,1,4}$, such that $d_p(f, \mathsf{w}) \le \epsilon$.

The sketch of the proof is simple: since we have required the transformation pair to be invertible and exact, so for any seq-to-seq function, we can universally approximate it in the wavelet space and it is equivalent to having universal approximation power in the original space. The detailed proof of Theorem 1 is shown in appendix A.1.

## 3 EXPERIMENTS

Besides intuitively and theoretically showing the superiority of the WISE paradigm, we also empirically show the accuracy of our design when compared with other recent efficient transformers, along with several ablations on each component of our Waveformer.

### 3.1 EXPERIMENTAL DESIGN

To examine the performance of various efficient transformer variants, we compare them on datasets that require long-range understandings, (1) five datasets from the publicly available benchmark Long Range Arena, and (2) two code understanding datasets.

**Long Range Arena (LRA)** Tay et al. (2020b) is a recent benchmark that is designed to compare different efficient transformers. Since its release which already contains ten different efficient transformers, more and more efficient transformers have chosen it as the primary evaluation. The benchmark contains five datasets to evaluate[1]. The datasets require understanding long sequences of mathematical operations, classifying text based on sentiment, matching similar documents, classifying images, and recognizing 2D spacial information. The sequence lengths of the dataset are within the range of 1K-4K.

**Code Understanding Tasks** Apart from the rather synthetic datasets in LRA (e.g., flattening CIFAR-10 images to simulate long sequence lengths), we also select a real-world long sequence task that involves detecting the vulnerabilities in source code. This task test Waveformer's ability to learn on source code and classify a piece of source code as buggy or not. Two datasets are included: CodeXGLUE (Lu et al., 2021) is introduced as a multi-programming language benchmark for code

---

[1]There are also hard variants that even full attention transformers fail to solve, which is out-of-scope of this paper.

Table 2: Sequence length for each dataset's train split. The task of vulnerability detection requires reasoning over the entire piece of code snippet to deduct the label.

| Dataset | CodeXGLUE | D2A-Func |
|---|---|---|
| Average Length | 1277.49 | 1038.86 |
| Median Length | 561 | 717 |

Table 3: Accuracy on vulnerability detection tasks. [†] Evaluation on validation set.

| Model | CodeXGLUE | D2A-Func |
|---|---|---|
| Code/C BERT | 62.08 | 60.2 |
| BERT-medium | 59.69 | 59.73[†] |
| Waveformer | 62.81 | 62.58[†] |

model evaluation, one of the tasks is vulnerability detection on C code, and the labels are created by security experts; D2A (Zheng et al., 2021)[2] is created for vulnerability detection with multi-modal trace and function data, we only use the function subset for our evaluation. To show the long-range nature of these tasks, we show the average sequence length (in the number of tokens after tokenization) in Table 2.

**Experiment Environment**. Our early-stage experiments are mostly conducted on RTX 3090 GPUs, and later moved to TPU v3-8s. Our code is written in Jax (Bradbury et al., 2018) with the Flax framework (Heek et al., 2020). The wavelet transformation implementation is primarily based on Jax Wavelet Toolbox (Moritz Wolter, 2021) and PyWavelets (Lee et al., 2019).

## 3.2 WAVEFORMER

**LRA**. We first highlight the performance of our method on the LRA benchmark in Table 1. Notably, among all the attention approximation methods, our Waveformer performed the best on three of the five datasets, and with a close-to-top performance on another. The time complexity of Waveformer is also the theoretically lowest possible among all other approximation methods, matching the size of the input.

**Code Understanding**. To show that our Waveformer's long-range reasoning ability can also be applied to real-world tasks, we compare it with a full attention BERT model on solving the two code understanding tasks. For Waveformer and BERT, we use a BERT-medium configuration (8 layers, 8 heads, 512 hidden dimensions) as the model size due to resource constraints.

We mostly follow existing works, Code-BERT (Feng et al., 2020) and C-BERT (Buratti et al., 2020), on the training and evaluation settings on the two datasets. Following Code-BERT's practice, we pre-train on CodeSearchNet (Husain et al., 2019) corpus that contains 6.4M code snippets across six programming languages (Python, Java, JavaScript, PHP, Ruby, and Go) with the masked language modeling objective. We then finetune our models on the downstream vulnerability detection datasets. Both Code-BERT and C-BERT take in sequence up to 512 tokens, but the code snippets' median length already surpassed 512, meaning that more than half of the inputs will be truncated. Therefore we set the max sequence length as 1,024 for our Waveformer and BERT. We also note that there is more extensive training for Code-BERT and C-BERT that we could not afford, for instance, Code-BERT's pre-training uses batch size 2,048 over 100K training steps whereas our batch size is set to 64, effectively 32 times less training epochs, and both models choose a 12-layer 768-dimension configuration compared to our 8-layer 512-dimension. The detailed hyperparameters and training configurations are included in Appendix A.4. We report their published test accuracy for CodeXGLUE and D2A-Func respectively as a strong baseline.

Table 3 contains the performance of Waveformer and BERT-medium on the two code-understanding tasks. We can see that on a fairground of training and model size, Waveformer outperforms a full-attention transformer model by a large margin. Further, when compared with larger-sized models that have dedicated training methodologies, Waveformer performs on par on the vulnerability detection task of CodeXGLUE, due to the power of abling to handle long-range sequences and understanding long-range relationships.

## 3.3 ATTENTION IN WISE

Our WISE paradigm has a general philosophy of applying non-linearity in the wavelet transformed space and is not limited to a certain type of attention method. We comprehensively evaluate rep-

---

[2]We note that D2A-Func hasn't released the test set, hence we report the validation accuracy for our implementations instead.

Table 4: We use $\mathcal{F}/\mathcal{W}$ to denote that the attention is performed in the Fourier/wavelet space (which also incurs an $O(n \log n)/O(n)$ complexity cost). [†] We reran Linformer & Linear Trans. for all (N/A, $\mathcal{F}$, $\mathcal{W}$) with the same additional five sets of hyperparameters because of convergence issues.[‡] We note that we are unable to reproduce a score close to the original Linformer performance on Pathfinder. [§] This is the normalized version of Performer as described in Section 2.3, when combined with the wavelet space, it is our Waveformer.

| Attention | | ListOps | | | Text | | | Retrieval | | | Image | | | Pathfinder | | |
|---|---|---|---|---|---|---|---|---|---|---|---|---|---|---|---|---|
| | | N/A | $\mathcal{F}$ | $\mathcal{W}$ | N/A | $\mathcal{F}$ | $\mathcal{W}$ | N/A | $\mathcal{F}$ | $\mathcal{W}$ | N/A | $\mathcal{F}$ | $\mathcal{W}$ | N/A | $\mathcal{F}$ | $\mathcal{W}$ |
| Full | $O(n^2)$ | 36.37 | 17.80 | 37.15 | 64.27 | 56.42 | 74.82 | 57.46 | 51.78 | 72.43 | 42.44 | 31.41 | 42.29 | 71.40 | 50.55 | 78.25 |
| Linformer | $O(n)$ | 35.70 | 36.15 | 37.65 | 53.94 | 57.06 | 55.22 | 52.27 | 55.93 | 65.85 | 38.47[†] | 34.89[†] | 39.17[†] | 66.44[†][‡] | 61.76[†] | 70.21[†] |
| Linear Att. | $O(n)$ | 16.13 | 37.65 | 37.55 | 65.90 | 71.66 | 71.93 | 53.09 | 72.71 | 70.71 | 42.32[†] | 51.07[†] | 40.83[†] | 75.91[†] | 70.45[†] | 76.43[†] |
| Longformer | $O(n)$ | 35.63 | 18.95 | 36.65 | 62.85 | 55.36 | 74.99 | 56.89 | 52.52 | 66.21 | 42.22 | 29.12 | 37.10 | 49.71 | 50.38 | 78.15 |
| Performer[§] | $O(n)$ | 18.01 | 37.15 | 38.20 | 65.40 | 65.52 | 75.60 | 53.82 | 60.56 | 78.56 | 42.77 | 9.99 | 42.98 | 77.05 | 50.49 | 79.17 |

resentative attention methods on different space transformations (no transformation, Fourier transformation, and wavelet transformation). We show that performing full attention, or many other attention approximation operations in a wavelet transformed space as proposed in WISE paradigm almost always brings great accuracy improvements. In Table 4 almost all attention methods have increased accuracy when applied in the wavelet space compared to an untransformed space, except for the Image dataset, where some methods incur a slight drop in accuracy. When compared with Fourier transformation, we can see that in most cases, the wavelet transformation is much better.

We do note an interesting finding that Linear Trans. when coupled with a Fourier transformation space, showed pretty good results on all the tasks, and especially a significant improvement on the Image task[3]. One hypothesis is that this is due to the fact that Linear Trans.'s polynomial kernel in Fourier space, which occurs in the forms of $K(x, y) = (x^\top y)$, can be interpreted as self-convolution in input space. We leave this for future discovery.

## 4 RELATED WORK

### 4.1 ATTENTION APPROXIMATION METHODS

There has been plenty of prior work to enable transformers to handle long input more efficiently and effective. Since the inefficiency comes from the quadratic dependency on sequence length because of the dense attention operation, a large portion of research simulates the attention operation with certain approximations, for example, replacing the dense attention matrix with a sparse version, or assume that it satisfies certain low-rank structures. We briefly review some methods on this topic in this section. For a more detailed survey, we refer the readers to Tay et al. (2020c).

**Sparse Attention**. Perhaps the most intuitive solution to alleviate the quadratic cost, Sparse Attention only calculates a portion of the full $n^2$ attention matrix. Early stage methods include Local Attention (Parmar et al., 2018) and Multi-passage BERT (Wang et al., 2019) use sliding windows or chunked blocks to speed up computation. Longformer (Beltagy et al., 2020) and BigBird (Zaheer et al., 2020) further combine global attention, sliding window attention, dilated sliding window attention, and random attention together to form strong sparse attention mechanisms, and BigBird showed that their method is a universal approximator of sequence functions. To make the block truncation a learnable process, Reformer (Kitaev et al., 2019) groups and sorts input segments via locality-sensitive hashing such that similar tokens are placed in the same chunk. Similarly, Sinkhorn Transformer (Tay et al., 2020a) trains a meta sorting network to reorganize input sequences before applying windowed attention.

**Low-rank Approximation**. The self-attention matrix, at the center of transformer, has been found to display low-rank behaviors after pre-training. Linformer (Wang et al., 2020) performed spectrum analysis on the pre-trained attention matrix, and the results indicate that the top 128 singular values composite 88%-96% of the entire 512 singular values across attention heads and layers. Based on this observation, Linformer added low-rank projection matrices in attention to approximate the original attention matrix. On a similar notion, Drone (Chen et al., 2021b) extended the low-rank approximation scope to all matrices in transformer via data-driven optimal compression.

---

[3]Note that the hyperparameters were tuned for Linear Trans. on this task, see Appendix A.3

**Kernel Methods.** The kernel methods approximate the whole self-attention by replacing the softmax with a kernel function that can be decomposed to avoid the explicit calculation of the $O(n^2)$ matrix multiplication. Linear Transformer (Katharopoulos et al., 2020) proposed a non-negative $\mathrm{elu}$ feature mapping as the substitution for the softmax, they further pointed out the connection between their formulation and RNNs, and argued that transformers and RNNs can be unified under the same umbrella. Building on top of this, Random Feature Attention (Peng et al., 2020) and Performer (Choromanski et al., 2020) utilized random feature approximation of the attention, one highlights the importance of normalization before random projection while the other one emphasizes the benefits of positive & orthogonal random features.

**Token Mixing.** Token Mixing methods are another version of efficient transformer building blocks. Different from the methods discussed above, they do not approximate attention, but rather conduct a new way of enabling communication between tokens. You et al. (2020) showed the possibility that a random token mixing strategy can work well in transformer encoders, as opposed to delicate (pre-)trained attention heads. Token Mixing is a new view towards self-attention as methods are not approximating self-attention. Lee-Thorp et al. (2021) pushed this idea further by providing an efficient method to mix the tokens with Fourier forward transformation.

Among these methods, our Waveformer utilizes a wavelet transform, thus, is slightly similar to Token Mixing. However, our work should be seen as a new approach to efficient transformers, which mixes the idea of a orthogonal space transform that communicates between tokens and attention approximations methods that can benefit in the new space. In our study, we also pick representatives from each of the of attention approximation types and show that the dense attention operation and these sparse attention operations can benefit from learning in a wavelet transformed space.

## 4.2 FOURIER & WAVELET TRANSFORM IN ML

Fast Fourier Transform (Cooley & Tukey, 1965) has been widely used in many machine learning domains, probably the most common usage is to speed up the computation of convolution. For a similar purpose to our work, AFNO (Guibas et al., 2021) learns global filters for images via adding block-wise MLP in Fourier transformation, equivalent to a convolutional layer with large filters. However, AFNO is designed for visual inputs, we show in our ablation study that such architecture cannot fully capture long-range text information.

Fast Wavelet Transform (Mallat, 1989) has been the backbone of numerous modern technologies including JPEG-2000 image compression (Skodras et al., 2001), digital communication (Akansu & Smith, 1995) and many others. FWT has been used for computation speed up (Wolter et al., 2020), speech recognition (Tufekci & Gowdy, 2000), and time series analysis (Michau et al., 2022). Recently in computer vision, WaveMix (Jeevan & Sethi, 2022) proposed to mix the input images with forward wavelet transform. We note that our work differs from theirs by learning the non-linear transformation in the coefficient space amid forward and backward wavelet transform.

## 5 CONCLUSIONS

In this paper, we propose to learn sequential mapping in the wavelet coefficient space. Specifically, the inputs are first forward transformed into the wavelet space, then the sequential mapping is learned and finally, we reconstruct the transformed sequence back in the input space. We combine this paradigm with random feature kernels and propose Waveformer, that has universal approximation power and linear time complexity. When coupled with attention approximation methods, WISE can boost their performance on long-range understanding tasks while enjoying no extra cost in time complexity. Our experiments support the superiority of Waveformer and obtain strong performance on the LRA benchmark and code understanding datasets.

Through this work we have focused on performing attention in the transformed space, but how will other seq-to-seq models work in Fourier or wavelet space remains unknown. We also note that there exist hundreds of wavelet bases, each with different design purposes and properties. Findings crucial properties for conducting effective learning in the coefficient space and constructing adaptive wavelet basis accordingly are both interesting problems we leave for future work.

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

## A  APPENDIX

### A.1  PROOF FOR THEOREM 1

We define the function class $\mathcal{F}$ to be the set of all countinous functions that map a compact domain in $\mathbb{R}^{n \times d}$ to $\mathbb{R}^{n \times d}$.

We start from making the connection between random feature kernel and regular transformer block:

**Lemma 1**  *(Asymptotic Result for FAVOR+) The following is true for independent random $w_i$,*

$$\text{MSE}(\hat{\text{SM}}(x,y)) = \frac{1}{m} \exp\left(\|x+y\|^2\right) \text{SM}^2(x,y)(1 - \exp\left(-\|x+y\|^2\right))$$

$$\Rightarrow \lim_{\text{SM}(x,y) \to 0} \text{MSE}(\hat{\text{SM}}(x,y)) \to 0$$

*where* $\text{SM}$ *denotes the softmax kernel,* $\hat{\text{SM}}$ *denotes the random feature kernel, and* $\text{MSE}$ *stands for mean-squared error.*

The proof of this lemma can be found at Choromanski et al. (2020, Lemma 2). It tells us the the MSE error is upper bounded to a constant since $x$, $y$ is normalized beforehand, and vanishes to 0 as the original softmax kernel value tends to 0 and the number of random features $m$ tends to $+\infty$.

Next we use the main theorem of Yun et al. (2019). We denote the transformer network class that has positional encoding, $h$ heads, head size $s$, and hidden dimension $r$ as $\mathcal{T}^{h,s,r}$.

| Hyperparameter | $\text{Config}_1$ | $\text{Config}_2$ | $\text{Config}_3$ | $\text{Config}_4$ | $\text{Config}_5$ |
|---|---|---|---|---|---|
| **Layers** | 1 | 1 | 2 | 2 | 2 |
| **Embedding Dim.** | 128 | 128 | 128 | 256 | 256 |
| **Attention Dim.** | 64 | 64 | 64 | 64 | 64 |
| **MLP Dim.** | 128 | 128 | 256 | 1024 | 512 |
| **Attention Heads** | 8 | 8 | 2 | 4 | 4 |
| **Dropout** | 0.2 | 0.1 | 0.1 | 0.1 | 0.2 |
| **Attention Dropout** | 0.1 | 0.1 | 0.1 | 0.1 | 0.1 |

Table 5: Additional hyperparameter configurations tried for Linformer and Linear Trans. in Image and Pathfinder

**Lemma 2** $\forall p \in [1, +\infty)$, $\epsilon > 0$, and for any $f \in \mathcal{F}$, we can find a Transformer network $g \in \mathcal{T}^{2,1,4}$, such that $d_p(f, g) \le \epsilon$.

The proof of Lemma 2 constitutes of several steps, of which the first step is to approximate any function $f \in \mathcal{F}$ as a piece-wise constant function $\tilde{f}$. Since $f$ is continuous, the piece-wise constant approximation can be of arbitrary accuracy. Next they find a modified transformer $\tilde{g}$ with hardmax operator and a special class of activations. Finally they show that the transformer block $g$ is able to approximate $g$. The functional distance is then bounded by:

$$d_p(f, g) \le d_p(f, \tilde{f}) + d_p(\tilde{f}, \tilde{g}) + d_p(\tilde{g}, g) \le \epsilon$$

We show that with slight modification, the proof will work for Waveformer, and can be generalized to the WISE paradigm under certain constraints.

The proof is outlined below: For $\forall f \in \mathcal{F}$, its wavelet transform $\hat{f}$ (we will also use $f_w$ to denote this, see (7) for details) is still continuous. Hence, the discretization claim remains valid. We can then effectively approximate the self-attention transformer block with the FAVOR+ block up to $\frac{\epsilon}{4}$ difference by controlling the number of random features $m$. In the end, the backward reconstruction is exact, the distance bound becomes when we control the other three terms to be less than $\frac{1}{4}\epsilon$ as well:

$$d_p(f, \mathsf{w}) \le d_p(f_w, \tilde{f}_w) + d_p(\tilde{f}_w, \tilde{g}) + d_p(\tilde{g}, g) + d_p(g, \mathsf{w}) \le \epsilon \qquad \blacksquare$$

## A.2 Fourier Transform Pair

Given function $f(x), x \in \mathbb{R}$, the Fourier transform pair is defined as the following:

$$\text{Forward Fourier}: \quad \hat{f}(\omega) = \int_{-\infty}^{\infty} f(x) e^{-i2\pi\omega x} \, \mathrm{d}x \qquad (14)$$

$$\text{Backward Fourier}: \quad f(x) = \int_{-\infty}^{\infty} \hat{f}(\omega) e^{i2\pi\omega x} \, \mathrm{d}\omega \qquad (15)$$

where $\omega$ stands for frequency in Fourier space.

## A.3 LRA Configuration Details

We tried to follow all hyperparameters as suggested for each of the attention approximations with exceptions on Linformer and Linear Trans. in Image and Pathfinder. For them, we experimented with five additional configurations as shown in 5.

For all wavelet transform conducted in this work, we use Daubechies 2 (Daubechies, 1992) as the basis and we set level of decomposition to 1.

For Waveformer, the number of random features in random feature kernel is set as 256 for all text tasks (ListOps, Text, Retrivial), 512 for all image tasks (Image, Pathfinder).

Table 6: Ablation study on Long-Range Arena benchmark.

| Model | ListOps | Text | Retrieval | Image | Pathfinder | Avg | Avg (w/r) |
|---|---|---|---|---|---|---|---|
| Waveformer | 38.20 | 75.60 | 78.56 | 42.98 | 79.17 | 62.90 | 58.99 |
| Linear | 37.70 | 55.36 | 55.27 | 15.75 | 50.58 | 42.93 | 39.84 |
| Fourier | 36.85 | 65.52 | 60.56 | 9.99 | 50.49 | 44.68 | 40.71 |
| Forward Fourier | 37.15 | 64.91 | 65.98 | 37.84 | 53.39 | 51.85 | 48.32 |

## A.4 CODE PRE-TRAINING & FINETUNING DETAILS

For pre-training, we use the CodeSearchNet dataset retrieved from Huggingface dataset. We use GPT-2 (Radford et al., 2019) tokenizer for Waveformer and BERT since source code does not have fixed vocabulary. We believe byte-level tokenization is more reasonable than sub-word tokenization.

We use the following set of hyper-parameters in pre-training: batch size is 64, learning rate is 5e-5, masking probability is 0.15, warm up steps is 10k, training steps is 100k with a linear learning schedule; we use AdamW optimizer with $\beta_1 = 0.9$, $\beta_2 = 0.98$, weight decay is 0.1.

We use the following set of hyper-parameters in finetuning: batch size is 64, learning rate is 2e-5, training length is 10 epochs; we use AdamW optimizer with $\beta_1 = 0.9, \beta_2 = 0.999$.

## A.5 ABLATION STUDY

We conduct an ablation study for Waveformer, as shown in Table 6. For (Linear), We limit the transformation in wavelet space to be linear. For (Fourier), we use the Fourier transform as the transformation mechanism for Waveformer. For (Forward Fourier), we only use the forward Fourier transform without backward transform. It can be observed that performance dropped significantly in all cases, indicating the necessity of non-linearity in wavelet space and forward-backward wavelet transform.

