# OpenReview forum: "Waveformer: Linear-Time Attention with Forward and Backward Wavelet Transform"
_ICLR.cc/2023/Conference — Submitted to ICLR 2023_

### Official Review · Reviewer_jCxJ · 2022-10-22

**Confidence:** 4
**Correctness:** 3
**Technical Novelty And Significance:** 2
**Empirical Novelty And Significance:** 1
**Recommendation:** 3

**Clarity, Quality, Novelty And Reproducibility:**

The paper writing is generally clear. I have questions regarding the novelty and the significance of the experimental results.

**Strength And Weaknesses:**

Strength:

1. The architecture is new as far as I know and studying the efficiency of the model is an important direction.
2. It is useful to show the model is a universal approximator, which is the same as the original Transformer model.

Weakness:

[Novelty]
1. Where is the linear complexity coming from?

If I understand correctly, the operation that reduces the quadratic complexity in attention is to use the random feature kernel developed in Choromanski et at., 2020, but not the proposed wavelet transform.

In the first round of reading, I got confused as in several parts of the paper, the authors emphasize that the linear-complexity model is a major contribution. I believe the authors have overclaimed that.

2. Why you use wavelet transform?

In the work, the input will first project to the coefficient space using a forward discrete wavelet transform. Then a linearized attention operation is applied. Note that even if you don;t use the wavelet transform, the model is already a linear transformer with random features. So is the wavelet transform redundant and what is the benefit?

[Universal approximation]

Once you use the random features, your model is, in fact, a "randomized" function. How did you deal with approximating a deterministic function by a "randomized function"? I didn't find any place in the proof coping with that. That being said, the proof is incomplete and lacks sufficient details. I don't think you can simply let the random feature dimension approach infinity and say, "MSE error vanishes to 0"， since we are concern about a finite-size Transformer's approximation ability

[Experiments]

The author missed important effcient Transformer S4 ("Efficiently Modeling Long Sequences with Structured State Spaces"). In all of the LRA tasks, S4 is roughly better than the proposed model for 20 points. I have little reason to vote a paper for acceptance given the significant performance gap.



**Summary Of The Paper:**

In the paper, the author proposed a new Transformer variant with linear complexity, which is called Waveformer. In each Waveformer layer, the input will first project to the coefficient space using a forward discrete wavelet transform. Then a linearized attention operation is applied via random features. Finally, the outputs are transformed back to the "contextual" space using a backward discrete wavelet transformer step. The authors studied the expressiveness of the architecture and conducted experimental analysis of the model on standard benchmarks.

**Summary Of The Review:**

I believe proposing efficient Transformer is essential but the current quality of the work (motivation, empirical results) are not ready to publish in the venue, and I recommend rejection.

---

> ### Author Response · Authors · 2022-11-17
> **Response to Reviewer jCxJ (R4)**
>
> We thank reviewer jCxJ for reviewing our manuscript. We answer the reviewer’s concern as follows:
>
> ***Where is the linear complexity coming from?*** - The reviewer is right about it that the linear attention mechanism came from performer, but our work is orthogonal to the attention methods. We mainly provide a framework (WISE) to learn on the wavelet basis, at a linear cost, as compared to FNet which costs O(n logn).
>
> ***Why you use wavelet transform?*** - The reviewer might have mistakenly understood our work. The benefit of learning in wavelet basis is explicitly shown in Section 3.3 and mostly in Table 4, where a comprehensive comparison has been made between learning in original space vs. Fourier space vs. wavelet space. Doing attention in wavelet space can boost the performance of a wide range of attention methods. We kindly ask the reviewer to check more carefully.
>
> ***Universal approximation*** - The reviewer questions the completeness of our universal approximation proof. We want to clarify that the main body of the random kernel proof came from performer paper, we added the normalization operation and extended the original proof, hence we didn’t provide more explanations.  Approximating deterministic functions with random features is fairly common in ML, for example, random feature kernel was proposed in 2007. The main idea mostly came from the Johnson-Lindenstrauss lemma, which states it is possible to preserve the distance between points after randomly projecting high-dimensional points to a low-dimensional space. We hope this answers the reviewer's question.
>
> ***Lack of Comparison with S4*** - We agree that S4 reached a new level of performance on LRA. However, firstly our work is orthogonal to S4 since we are studying the effect of conducting attention in transformed space - wavelet space; secondly S4 is a state space model that is vastly different from our transformer-based architectures. The comparison of our work should be made between methods like FNet. Our major contribution should be interpreted as wavelet space is better than the original space and Fourier space. This insight is novel and impactful. Chasing the best possible number is great but our insight is also valuable to the community.
>
> We hope this response clarifies the reviewer’s questions.

---

> > ### Comment · Reviewer_jCxJ · 2022-11-18
> > **Thanks for the reply**
> >
> > [Linear complexity]
> >
> > Thanks for the clarification. It is glad to know that the efficiency gain comes from the performer but not the wavelet transformation. It makes me confused that you highlight the "linear complexity" in the title and through the manuscript, as the benefit should be credited to another work
> >
> > [Performance comparison v.s. S4]
> >
> > I totally agree that this work makes modifications to the Transformer model. And comparisons between this work and Transformer variants are essential. However, you must evaluate this work and compare it with the SoTA. Everyone hopes his/her work is used by the community, but if the work is far behind the SoTA, why should others use it?

---

> > > ### Comment · Reviewer_ciFm · 2022-11-21
> > > **We shouldn't be too SoTA driven**
> > >
> > > I cannot agree with the reviewer's comment:
> > >
> > > "if the work is far behind the SoTA, why should others use it?"
> > >
> > > There are many types of research works. One type is to build a system that works well on benchmarks, i.e., achieving SoTA. The other is to study a core unit in a large system and gain deeper understanding. Each type has its own value. This work belongs to the second type. It is unfair to place it against works in the first type.

---

> > > > ### Author Response · Authors · 2022-11-21
> > > > **Thank you, Reviewer ciFm!**
> > > >
> > > > We believe our insights about using different transformed spaces and why Wavelet Transform stands out will be valuable to the community.

---

> > > > > ### Comment · Reviewer_jCxJ · 2022-11-21
> > > > > **Further response**
> > > > >
> > > > > Yes. SoTA-driven research is not the only way, and many works that provide a deep understanding of this field don't achieve SoTA. However, this work targets to solve a purely practical problem (to accelerate the Transformer blocks), and your work achieved significantly worse performance compared to a previous method with a similar inference speed (O(n) in your method v.s. O(nlogn) in S4).
> > > > >
> > > > > I saw there is no new revision to the paper, and you never tried to compare a previous work I refered to(S4) with your method. Then I ask one more question: for a practical problem we face, we have two choices to use: S4 and Waveformer. Which one would you recommend me to use and why?

---

> > > > ### Comment · Reviewer_jCxJ · 2022-11-21
> > > > **Apologize**
> > > >
> > > > I agree we shouldn't be too SoTA driven. But we should compare our work with published results in the past and fairly tell the audience the benefit of the proposed method compared to those works.
> > > >
> > > > Long sequence modeling is a particularly narrow direction focusing on developing efficient modeling methods for long sequence problems. I am trying to understand why the authors refuse to compare the proposed method with related works. The argument from the author seems to be "my work is on efficient Transformer. For any models other than Transformer, even if they are much better than my method, I will skip to discuss and compare with them."
> > > >
> > > > I am not satisfied with the arguments. But I am open to discuss on this with all reviewers.

---

> > > > > ### Comment · Reviewer_ciFm · 2022-11-21
> > > > > **We have to define the relevance in a proper way**
> > > > >
> > > > > I understand the reviewer's comment. Indeed the experimental section works on the same task as the S4 paper. However, other than that, I cannot find any other relevance.
> > > > >
> > > > > This paper mainly studies attention mechanism in transformer architecture. S4, on the other hand, suggests that in some scenarios, there could be a more effective alternative to transformers.
> > > > >
> > > > > Given these facts, I think an appropriate way is to clearly mention the performance gap with S4, and highlight to readers, that the goal is not to beat all SoTA, rather, the aim is to show the gain within the regime of attention and transformer architecture.

---

> > > > > ### Author Response · Authors · 2022-11-21
> > > > > **Proposed Change**
> > > > >
> > > > > On top of our current paper, we have shown more results with different wavelet bases & decomposition levels (see our response to Reviewer DaL7), varying time/frequency resolution (to further investigate why learning in wavelet basis is helpful, see our response to Reviewer ciFm), and more runs (see our response to Reviewer KfzV).
> > > > >
> > > > > As for S4, we propose to discuss and include it in related works, we'll elaborate on the performance gap and S4's strengths and weaknesses. Again, we want to clarify that our main contribution is not a single SOTA model (we used a modified performer as an example), but rather a learning paradigm that can boost performance for a wide range of attention methods. Hence we shall not be crucified for not beating all the other work that targets LRA.

---

> > > ### Author Response · Authors · 2022-11-21
> > > **Wavelet Transform is also important to the linear complexity**
> > >
> > > Re: linear complexity.
> > >
> > > Wavelet Transform also plays a role here --- we can only achieve the linear time complexity when the transformation itself can be done in a linear time. If one choses Fourier Transform, there is no way to achieve the linear time complexity because the fastest Fourier Transform we know requires O(NlogN). In contrast, Fast Wavelet Transform can be done in O(N).

---

### Official Review · Reviewer_ciFm · 2022-10-22

**Confidence:** 4
**Correctness:** 3
**Technical Novelty And Significance:** 3
**Empirical Novelty And Significance:** 3
**Recommendation:** 6

**Clarity, Quality, Novelty And Reproducibility:**

Overall, the paper is well written. A few clarity questions:

1. Second to last paragraph in section 2.2: "... perform discrete wavelet transform over each spatial dimension of the input ..."--- by "spatial dimension" -- you mean each of the $d$ dimensions of the input feature?

2. The wavelet transform will keep time axis while adding another frequency axis. How is that handled? 2D input becomes 3D, or simply "flatten"/"reshape" to 2D?



**Strength And Weaknesses:**

The method is simple and interesting. Empirical results (especially Tab. 4) are encouraging. It encourages reader to think "
why attention in (wavelet) transformation domain can outperform the conventional one in time domain?".

However, the answer is still unknown after reading the paper. An in-depth analysis is lacking. Section 2.2 discusses the advantage of wavelet coefficients against Fourier coefficients. The authors suggests maintaining both time and frequency resolution is helpful. But that is a signal processing perspective, whose connection to representations learning is not clear. In that sense, is short-time Fourier transform also able to outperform Fourier transform? Theorem 1 is a combination of two existing theoretical results. It still cannot provide an answer to the question above.

**Summary Of The Paper:**

The paper innovates attention block by sandwiching it into wavelet forward and backward transformations. Empirical results suggest that this approach can improve upon vanilla softmax attention, as well as other approximated and more efficient attention mechanisms.

**Summary Of The Review:**

Although an in-depth analysis and theoretical justification is lacking, the paper presents a simple method and promising results, which is novel and interesting.

---

> ### Author Response · Authors · 2022-11-17
> **Response to Reviewer ciFm (R3)**
>
> We thank the reviewer ciFm for providing helpful feedback! We respond to the reviewer’s comments below:
>
> ***In-depth analysis*** - The main concern of the reviewer is that we lack an in-depth analysis of “why attention in wavelet domain can perform better”. Indeed it is a crucial question, nonetheless a difficult one. We hypothesize that it is due to the wavelet’s balanced time-frequency resolution, further supported by a newly added experiment. We use STFT as the tool because its time-frequency resolution can be easily changed by changing its window size. We experimented with three cases - high time resolution (similar to original input space), balanced resolution (similar to wavelet space), and high frequency resolution (similar to Fourier space). The results are shown below (Waveformer’s test Acc is 75.6):
>
> Link to the graph: https://iili.io/H91AWMJ.png
>
> It can be observed that there is a trend going up then down when we go from one extreme to the other. But we agree that this is still much of an empirical study, figuring out what happens to the learned representation would be meaningful future work.
>
> ***Clarity***
> 1. *Do you mean each of the d dimensions of the input feature?* – Yes.
> 2. The new axis is handled as an independent axis to the feature axis and sequence length axis, i.e. we did not flatten over the resulting new axis.
>
> Again, we thank reviewer ciFm for your insight. We agree that there is a theoretical gap that lies in front of us, which we intend to leave for future work.

---

> > ### Comment · Reviewer_ciFm · 2022-11-22
> > **further clarify the dimension**
> >
> > The authors reply:
> > "The new axis is handled as an independent axis to the feature axis and sequence length axis, i.e. we did not flatten over the resulting new axis."
> >
> > However, my question isn't fully addressed. In specific, the traditional attention is computed between $T \times d$ query and key matrices. Now they both become 3D tensors, $T\times F \times d$, where $F$ is the number of frequency bins. How is the attention computed?

---

> > > ### Author Response · Authors · 2022-11-22
> > > **Explaining dimension**
> > >
> > > In reviewer ciFm's example, the attention is computed over the $T' \times d'$ matrices fixating each $F$ (note that the query and key matrices will have smaller size after transformation, $d>d', T>T'$). Hope this clarifies the question.

---

### Official Review · Reviewer_DaL7 · 2022-10-24

**Confidence:** 4
**Correctness:** 3
**Technical Novelty And Significance:** 2
**Empirical Novelty And Significance:** 3
**Recommendation:** 6

**Clarity, Quality, Novelty And Reproducibility:**

## Quality and clarity:

### Methods:
- Equation (1) previously appeared in the attention is all you need paper. A citation would be in order.
- Do equations 2, 4, 5, 7 and 8 appear in standard textbooks? Adding citations helps your readers find them.
- Figure 2: The third plot in figure 2 titled 'Wavelet Transform,' looks like it was produced by a continuous transform. Are the wavelet-transforms in the paper discrete or continuous?
- Section 2.2: The Short-Time-Fourier-Transform (STFT) is often used instead of wavelet-transforms to conserve temporal or spatial information. While the introduction to section 2.2 is true, not mentioning the STFT could be misleading.
- Fnet [Lee-Thorp et al.] uses two-dimensional Fourier-Transforms. Are the wavelet transforms used in this work two-dimensional?
- If a two-dimensional transform was used is the implementation using two 1D-transforms or a single two-dimensional transform?
- If 1D wavelets are used, how are these augmented to be useful in 2D?

### Experiments:
- Table 1: Are the experiments reproducible? Are seed values for all experiments known?
- Table 1: How big is the effect of different seeds? Do we know the variance on the numbers in table 1?
- Table 1: Is the performer experiment identical to the waveformer experiment if the analysis and synthesis wavelet-transforms are removed?
- Table 1: Are the results for the related work in table 1 reproductions or citations? If the values are cited a citation of the original paper should appear in the model column.
- Which wavelet was used for the experiments? How many coefficients does it have? How was it chosen?
- Which decomposition level do the transforms have? How was the level chosen?
- Table 4: Are accuracy values tabulated in table 4?

### Copy editing:
- The paper requires additional copy editing. Minor flaws are mostly related to articles and include i.e:
    - After the introduction:
        - The first line could be either "the transformer" or "transformers".
    - In the contribution section:
        - i.e. we develop the waveformer, which requires ...

## Originality:
- Wavelet transforms are well-known in machine learning. The papers-related work section addresses this fact properly. In the context of attention introducing wavelets is original.


**Strength And Weaknesses:**

### Strengths:
- To the best of my knowledge the fast wavelet transform has not yet been studied in attention modules.
- The theory is adequately discussed.
- The reported results are competitive.

### Weaknesses:
- The novelty of the proposed approach is limited. Wavelet transforms have previously been studied in other contexts.
- For a paper which studies the introduction of wavelet transforms, it would have been interesting to study the effect of the wavelet itself.
- It seems different wavelets have not been explored.
- Similarly, the depth of the wavelet transforms has not been studied. It would have been interesting to see if and how the level choice impacts the results.

**Summary Of The Paper:**

The paper "WAVEFORMER: LINEAR-TIME ATTENTION WITH FORWARD AND BACKWARD WAVELET TRANSFORM" proposes integrating analysis- and synthesis-wavelet transforms into transformers.

The main idea is to move attention computations into the wavelet domain, by framing the attention computation with a forward and backward wavelet transform. The attention module uses the randomized mapping known from the performer architecture.
The proposed network is evaluated on the Long Range Arena and the Code Understanding benchmarks.

The authors observe competitive or improved performance.

**Summary Of The Review:**

This paper presents an interesting application of fast wavelet transforms in attention modules.
The results are competitive. The experimental part of the paper leaves many questions open.
Wavelet choice and the number of scales for example are not investigated.
Mean, and variance are unknown for all experiments. Experimentally the paper presents the bare minimum.
I am recommending a weak reject for now, but am willing to adjust my rating depending on the rebuttal.

Edit: In response to the rebuttal, I have raised my rating from five to six.

---

> ### Author Response · Authors · 2022-11-17
> **Response to Reviewer DaL7 (R2)**
>
> We thank the reviewer for your insightful comments! Answering some of your questions below:
>
> ***Effect of wavelet choice & decomposition level*** - As the reviewer pointed out, we did not discuss the effect of wavelet basis and decomposition level. Here we add more empirical analysis for this, the key takeaway is that the basis itself does not matter too much, but the decomposition level provides a trade-off between text tasks and image tasks:
>
> |               | ListOPs | Text  | Retrivial | Image | Pathfinder |
> |---------------|---------|-------|-----------|-------|------------|
> | db3, L=1      |   37.85 | 76.86 |     73.62 |  42.3 |       78.3 |
> | db3, L=2      |    37.4 | 76.84 |     75.43 |  41.2 |      50.52 |
> | coiflet1, L=1 |   36.85 |  7672 |     75.43 | 41.42 |      50.58 |
> | coiflet1, L=2 |   37.65 | 75.97 |     75.29 |  43.2 |      77.49 |
> | symlet2, L=1  |    37.5 | 75.07 |     75.59 | 42.85 |      49.85 |
> | symlet2, L=2  |   37.45 | 75.63 |     74.51 | 41.03 |      77.39 |
> | symlet2, L=3  |   37.55 | 75.39 |     76.24 | 40.88 |      76.84 |
>
> ***Wavelet Implementation*** - All wavelet transform used in this paper is discrete, two-dimensional (done in a single two-dimensional transformation). We used Daubechies 2 wavelet as the basis, and the decomposition level is set to1. The choice was made because of db2 transform’s efficiency.
>
> ***Experiments*** - For reproducibility and stability of the results, please see our response to Reviewer KfzV (R1). Waveformer’s random feature kernel is different from performer by an additional normalization (see Section 2.3), the normalization matters because without it the MSE becomes unbounded. In table 1, the values are cited, and the values are accuracy in table 4.
>
> ***Miscellaneous*** - The suggested edits and citations are valuable, we will address them in the paper. And specifically for STFT, we added an analysis using it as the tool to show why mixing time and frequency helps, and as a comparison to wavelet. See our response to R3 - ciFm for details.
>
> We agree with the reviewer that exploring the connection between wavelets’ properties and the model output would be an interesting direction to explore in future work. However in this work, we are highlighting the importance and benefits of learning attention in wavelet space.
>
> We thank Reviewer DaL7 for their suggestions and comments, and we hope that the above responses adequately address all concerns.

---

> > ### Comment · Reviewer_DaL7 · 2022-11-21
> > **Thank you for the careful reply.**
> >
> > ### Methodology:
> > #### Figure 2:
> > Attempting to reproduce figure two led to the following code snippet:
> > ``` python
> > import numpy as np
> > import pywt
> > import scipy.signal
> > import matplotlib.pyplot as plt
> >
> > t = np.linspace(0, 10, 512); chirp = scipy.signal.chirp(t, 1, 10, 4);
> > # Discrete transform
> > plt.plot(np.concatenate(pywt.wavedec(chirp, 'db8'))); plt.title('discrete'); plt.show()
> > # Continuous transform
> > plt.imshow(np.abs(pywt.cwt(chirp, scales=np.arange(1,64), wavelet='morl')[0])); plt.title('continuous'); plt.show()
> > ```
> > It seems the continuous `cwt` function is required to obtain results which are similar those shown in figure two. Would it be wise to add a hint in the figure's caption to guide readers in the right direction? Given the answers to reviewer ciFm's questions, the CWT and STFT now appear in the paper. Will both be introduced briefly in the final version of this draft?
> >
> > ### Experiments
> > Thank you for running additional experiments. I believe that having the ability to choose from multiple wavelets is a crucial difference compared to other algorithms like the STFT. Readers now have the option to pick from the table. Furthermore, investigating the variance in response to KfzV was crucial.

---

> > > ### Author Response · Authors · 2022-11-21
> > > **Thanks for your suggestions.**
> > >
> > > Indeed, we did not mention that Figure 2 was made with the continuous wavelet transform. We'll cover that in the final draft as well as for STFT. Let us know if you have any other concerns, we'll be happy to address them.

---

> > > > ### Comment · Reviewer_DaL7 · 2022-11-24
> > > > **Rating revised:**
> > > >
> > > > The authors have:
> > > > - Presented experiments exploring different choices for the wavelet and the number of scales.
> > > > - Promised to help their readers find related work by adding citations for standard equations.
> > > > - Promised to discuss the variance for different seeds as suggested by reviewer KfzV.
> > > > - Promised to mention the cwt in the caption of figure two.
> > > > - Agreed to add a short discussion of the STFT to ensure readers are aware of this alternative method.
> > > >
> > > > I think the paper can be improved sufficiently in time for ICLR 2023. To the best of my knowledge, wavelet transforms have never been integrated into transformers before. Suppose the variance is adequately discussed in the final draft, as suggested by KfzV, alongside the numbers presented in the rebuttal to this review. In that case, I believe the paper can be accepted. I have adjusted my rating accordingly.

---

### Official Review · Reviewer_KfzV · 2022-10-27

**Confidence:** 3
**Correctness:** 3
**Technical Novelty And Significance:** 2
**Empirical Novelty And Significance:** 2
**Recommendation:** 5

**Clarity, Quality, Novelty And Reproducibility:**

Sufficient technical details and background knowledge are presented, though the authors may need to further justify the novelty of the proposed method.

**Strength And Weaknesses:**

Strengths:
* The authors focussed on an important issues in sequence modeling
* The paper is organized well and easy to follow
* Technical details are presented clearly

Weakness:
* The novelty needs to be further justified
* The evaluation needs to be strengthen

**Summary Of The Paper:**

The authors propose a wavelet transformation based liner attention mechanism named Waveformer. Experiments on long-range arena benchmarks have been conducted to demonstrate the effectiveness of the proposed method.

**Summary Of The Review:**

My major concern is the novelty of the proposed method. Applying spectral-based transformation (Fourier or wavelet) for efficient computation has been widely adopted in ML (also indicated in Sec. 4.2 by the authors). The claim of non-linear transformation as a major contribution seems to be incremental. I suggest the authors to better justify the novelty of the proposed approach and the motivation of applying wavelet transformation.

For experiments, I suggest the authors explicitly indicate how many repetitions each model has been trained and also report the variance of accuracy. The authors choose random feature kernel in the paper and I wonder if the authors have considered other kernels or how different kernels may affect the model performance.

---

> ### Author Response · Authors · 2022-11-17
> **Response to Reviewer KfzV (R1)**
>
> Thanks for your review! We are addressing some of your comments below:
>
> ***Novelty*** - Transformations such as Fourier transform indeed have been widely used in ML. **But our novelty came from the new finding that doing attention in wavelet space is much better than in the original input space or in the Fourier space**, for a wide range of attention methods (see Page 8 Table 4 and Section 3.3). This brings new insights to the community. And during the rebuttal period, we further investigated why learning in wavelet space performs better, the conclusion is that wavelet space provides both time and frequency resolution, hence learning there results in better performance (see our response to R3 - ciFM for details).
>
> ***Better Evaluation*** - We conducted three more runs with three different seeds for the LRA benchmark experiment. The mean and standard deviation are listed below:
>
> |       | ListOPs | Text  | Retrivial | Image | Pathfinder |
> |-------|---------|-------|-----------|-------|------------|
> | Mean  |   37.88 | 75.67 |     75.51 | 41.84 |      70.80 |
> | Var   |    0.30 |  0.40 |      2.22 |  0.81 |      13.77 |
>
> Note that the only one that had a large variance in the Pathfinder task, as it is inherently hard to train and sometimes collapse during training. We’ll integrate this into our paper.
>
> We again thank Reviewer KfzV for their review of our manuscript, and we hope that the above responses adequately address all concerns.

---

### Decision · Program_Chairs · 2023-01-20

**Decision:**

Reject

**Justification For Why Not Higher Score:**

The paper uses a linear transformation + kernel trick + single level of the simplest wavelet transform.
Should be the same as the kernel trick itself, but different initialization of the weights. A simple synthetic example is needed, and the writing of the paper could be improved.N

**Justification For Why Not Lower Score:**

N/A

**Metareview: Summary, Strengths And Weaknesses:**

The paper proposes to use two-dimensional fast wavelet transform in the attention module, and numerical comparison to numerous baseline are reported.

Strengths: This is the first attempt to use fast wavelet transform in the attention module, with improvement (sometime significant) reported for different tasks.

Weaknesses: The set of tasks considered in the paper is rather specific, and although in the discussion the authors say "we should not be crucified for not beating SOTA", the other side of this medal is "not reporting anything that is better than my thing", creating a false impression that they beat the result. But this is a minor issue. Another issue is that the type of the transform is not listed in the main text, but it should be important. The authors reported additional study in the rebuttal, claiming it is not important.
The paper is not well-written: it is not easy to recover the logic of the paper (what is applied to what, for example) by reading the text.

A more delicate issue is that the final transform they report to be using is DB2 (i.e. Haar) wavelet transform, which is very simple (half-sum, half difference) of the signal.  It is very difficult to believe that this result comes exclusively from some magic property of the wavelet transform. A simple theoretical example of sequence modeling is really needed to confirm that the method has better expressivity.



**Summary Of Ac-Reviewer Meeting:**

N/A